# PID-Based Longitudinal Control of Platooning Trucks

Aashish Shaju , Steve Southward and Mehdi Ahmadian *

Department of Mechanical Engineering, Virginia Tech, Blacksburg, VA 24061, USA; aashish00@vt.edu (A.S.); scsouth@exchange.vt.edu (S.S.)
* Correspondence: ahmadian@vt.edu

**Abstract:** This article focuses on the development and assessment of a PID-based computationally cost-efficient longitudinal control algorithm for platooning trucks. The study employs a linear controller with a nested architecture, wherein the inner loop regulates relative velocities while the outer loop governs inter-vehicle distances within platoon vehicles. The design of the proposed PID controller entails a comprehensive focus on system identification, particularly emphasizing actuation dynamics. The simulation framework used in this study has been established through the integration of TruckSim® and Simulink®, resulting in a co-simulation environment. Simulink® serves as the platform for control action implementation, while TruckSim® simulates the vehicle's dynamic behavior, thereby closely replicating real world conditions. The significant effort in fine-tuning the PID controller is described in detail, including the system identification of the linearized longitudinal dynamic model of the truck. The implementation is followed by an extensive series of simulation tests, systematically evaluating the controller's performance, stability, and robustness. The results verify the effectiveness of the proposed controller in various leading truck operational scenarios. Furthermore, the controller's robustness to large fluctuations in road grade and payload weight, which is commonly experienced in commercial vehicles, is evaluated. The simulation results indicate the controller's ability to compensate for changes in both road grade and payload. Additionally, an initial assessment of the controller's efficiency is conducted by comparing the commanded control efforts (total torque on wheels) along with the total fuel consumed. This initial analysis suggests that the controller exhibits minimal aggressive tendencies.

**Keywords:** truck platooning; longitudinal control; nested PID; TruckSim®; Simulink®; co-simulation

## 1. Introduction

The advent of autonomous vehicles has revolutionized the transportation industry, promising safer, more efficient, and sustainable mobility solutions. These self-driving vehicles, equipped with advanced sensors and artificial intelligence [1], have the potential to transform the way we travel and significantly impact various sectors, including logistics, public transportation, and personal commuting. One of the subdomains of autonomous vehicles is platooning, a technology that enables a group of vehicles to operate in close formation, enhancing traffic flow and fuel efficiency.

Truck platoons refer to multiple freight trucks that travel in a closely coordinated convoy (connected to each other through V2V communications), with one vehicle following another at a close distance. The concept of platooning is not entirely new; however, advancements in autonomous driving technologies have unlocked its true potential, making it a compelling solution for addressing contemporary transportation challenges. Expected advantages encompass reduced fuel consumption [2], optimized road capacity [3], and reduction in personnel costs [4]. Furthermore, the growing trend of vehicle electrification has renewed enthusiasm for platooning [5,6].

Numerous multidisciplinary studies have been conducted on autonomous, connected vehicles, spanning fields such as transportation engineering, computer science, control systems, communication networks [7,8], and urban planning [9–11]. These works focus on

aspects like vehicle-to-vehicle (V2V) [12] and vehicle-to-infrastructure (V2I) [13] communication, sensor integration, path planning, human–machine interaction, cybersecurity [14,15], control development, energy efficiency, traffic flow optimization, and policy implications.

Most studies in the control algorithm development domain rely on either kinematic vehicle models or a mixed kinematic, dynamic model, including aerodynamic drag, tire model, powertrain dynamics, etc. [16–20] Even in such fully non-linear models, acquiring all the necessary parameters for intricate longitudinal models, such as gearbox characteristics, remains a challenge due to inherent discontinuities (gear changes). Additionally, due to the real-time computational demands, the application of Model Predictive Control (MPC) using intricate models [21] becomes unfeasible. In this study, we try to address some of these concerns by devising a computationally efficient controller that maintains a comparable tracking performance to these non-linear controllers.

String stability is another important feature to be considered when the entire platoon is considered as a dynamic system. A criterion for string stability, proposed by Cremer [22], is based solely on the velocity error of each vehicle with respect to the leading vehicle's velocity. Moreover, Swaroop et al. [23] delineated string stability prerequisites classified into strong sense and weak sense, predicated on inter-vehicle distances. In practical scenarios, achieving string stability often involves maintaining a constant inter-vehicular spacing, as seen in the constant spacing policy [24], or permitting variation based on the ego vehicle's velocity, as exemplified by the constant time headway (CTH) policy [25].

The field has also seen extensive research employing Model Predictive Control [26,27] (MPC). Literature pertaining to distributed receding horizon control has investigated interconnected subsystem dynamics for both linear [28,29] and non-linear [30] system behaviors. Additionally, the domain has explored the application of distributed receding horizon control for multiple, decoupled vehicles, considering linear [31] and non-linear [32] vehicle dynamics, often incorporating coupling within cost functions and constraints. Moreover, various non-linear control strategies, notably Sliding Mode Control, have been investigated to enhance longitudinal control within automated platoons [16,18,24,25,33,34]. A notable limitation observed across many of these studies is their simulations being initialized with zero/small initial spacing errors, which raises practical concerns regarding their real-world applicability.

There are also various previous studies that have already explored the use of PID-based longitudinal controllers, both within platooning scenarios and in the broader context of autonomous vehicles [35,36]. The novelty of this research lies in the adoption of a nested architecture within the proposed controller. This approach offers distinct advantages in terms of decoupled control. By allowing the management of control signals for distinct control objectives independently, the nested structure mitigates cross-coupling effects and facilitates precise control over individual elements. Furthermore, the nested design offers benefits in terms of actuator saturation management. This feature provides the flexibility to establish independent saturation limits for each loop, thereby preventing issues related to integrator wind-up and enhancing the controller's capacity to navigate constraints on actuator commands. Additionally, the nested architecture demonstrates improved disturbance rejection capabilities. Through the attenuation of disturbances within inner loops prior to their propagation to the outer loop, superior disturbance rejection and smoother overall control are achieved.

In this paper, Section 2 outlines the comprehensive setup of the co-simulation framework and the system identification of the inverse actuators employed for generating actuation signals from the PID output. Section 3 provides an in-depth exploration of the controller design process, starting with the longitudinal model estimation and progressing to the application of loop shaping techniques, along with a brief analysis of the controller's stability. Subsequent Section 4 entails a thorough discussion of the acquired results, including their limitations, overall practicability, and applicability. Additionally, it highlights the distinctive aspects of the proposed algorithm in contrast to existing related studies. Finally,

Section 5 serves as the conclusion, encapsulating the contributions of this research and outlining potential future works.

## 2. TruckSim®-Simulink® Co-Simulation Framework

Figure 1 shows the implementation of the co-simulation framework developed for the study. It shows the communication paths between all the vehicles in the platoon, which in the real world would be established using V2V wireless communication. The illustration has been color-coded for better interpretation of the operating environment of each block (blue—Simulink® and green—TruckSim®). The output of the controller is the total torque required at the wheels, which can be both positive or negative depending upon the following vehicles' positions and velocities with respect to each other and the lead vehicle, as illustrated. The PID output is then mapped into respective actuation signals (throttle for positive torque and brake pressure for negative torque) using the inverse actuator dynamic models. These models have been identified through system identification, as discussed in subsequent sections. These resultant actuator inputs are then introduced into the TruckSim® simulation, which functions as the dynamic model for a tractor–trailer combination [37].

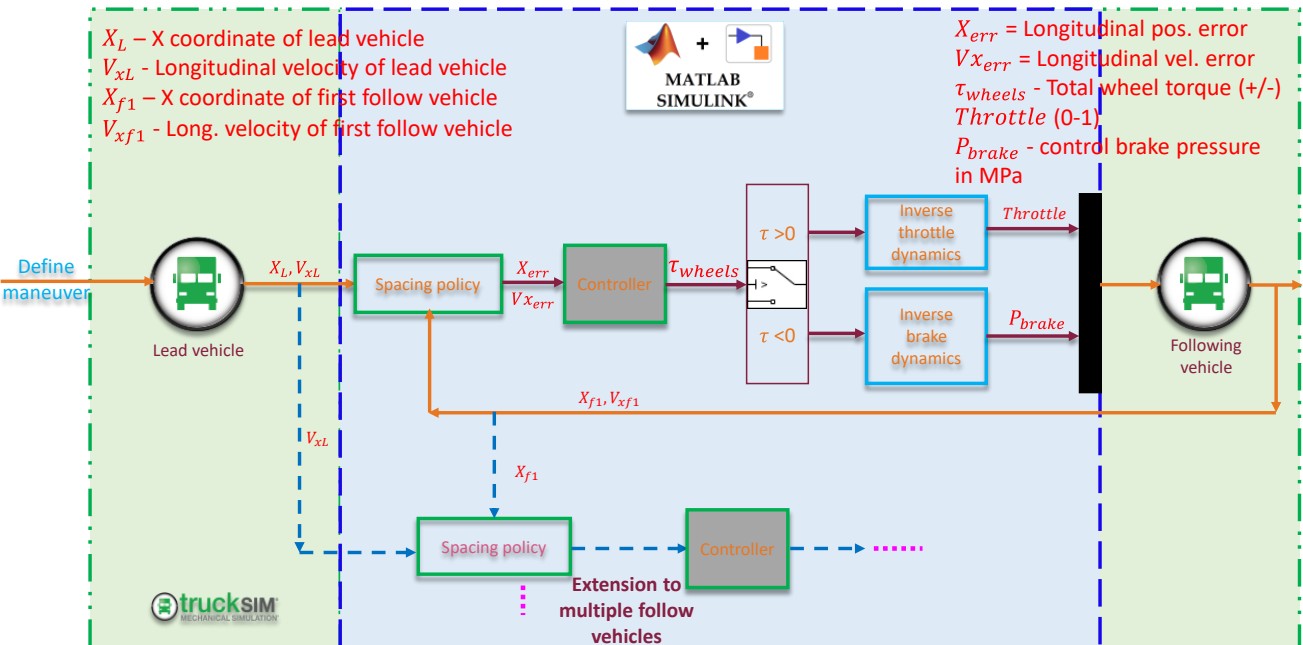

**Figure 1.** TruckSim®-Simulink® co-simulation framework.

The spacing strategy used in this work adopts the constant time headway (CTH) policy. CTH is widely utilized for its capability to enhance a platoon's string stability. According to the CTH policy:

$$D_i(t) = d_{min} + t_h v(t)_{follow} \tag{1}$$

where, $D_i(t)$ is the required space between vehicle $i$ and $i - 1$; $d_{min}$ is the standstill distance between vehicles $i$ and $i - 1$; $t_h$ is the headway time constant taken as 0.3 s; and $v(t)_{follow}$ is the ego vehicle velocity at time '$t$'.

**System Identification of the inverse actuator dynamics**

Figure 2 illustrates an expanded representation of the control signal transmission from Simulink® to TruckSim®. The application of inverse transfer functions $\left( \frac{1}{TF_{throttle}}, \frac{1}{TF_{brake}} \right)$ effectively nullifies the intrinsic powertrain and brake system dynamics of the vehicle. This configuration enables us to:

(a)    Employ a single PID controller for both acceleration and braking functions.
(b)    Fine-tune PID controller gains by shaping the control loops based on the linearized model of longitudinal dynamics.

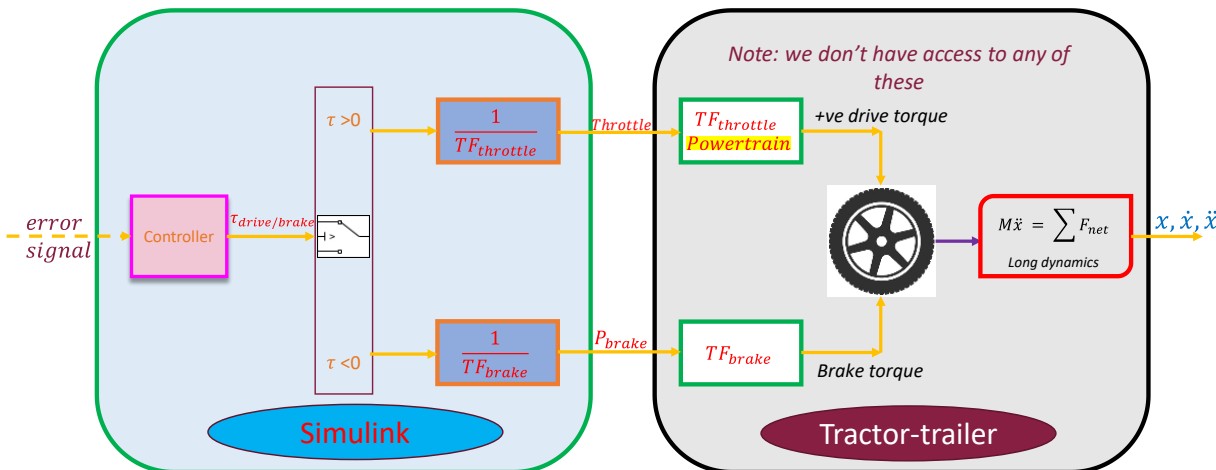

**Figure 2.** Block diagram of the implemented control and routing of the actuation signals.

### Characterization and validation of inverse throttle dynamics

A linear system identification approach was used to estimate the transfer function (defined as inverse throttle dynamics, refer to Figure 2) between the total torque at the drive axle/axles required (i/p) and the throttle value (o/p—a dimensionless quantity ranging from 0 to 1).

For system identification, it is essential to have a suitable set of input and output data. To estimate the inverse throttle dynamics (which relates total drive torque to the throttle input), we require drive torque data as the input dataset and the corresponding throttle data as the output dataset. However, TruckSim® does not allow the direct input of drive torque data into its simulation environment. Consequently, we designed a throttle profile (a series of step inputs since it gives a good understanding of the system's transient dynamics, steady state value, and stability) to serve as the input, measured the corresponding drive torque, and then rearranged this i/o dataset so as to treat drive torque as the input dataset and throttle as the output dataset for the subsequent transfer function estimation process.

### Zero transient filtering

The current study uses a single-drive axle day cab tractor as the driving unit, and as such, a total of seven estimated transfer functions (one for each gear) were estimated using the time domain system ID tools in MATLAB®. This resulted in a bank of different dynamic filter models corresponding to each gear. Consequently, when a gear change occurs, it necessitates the replacement of the current filter with one appropriate for the engaged gear from this set of filters. A common method of switching among filters using switches in Simulink® would generate unwanted transience in the output. This is because, as per the author's knowledge, the traditional ways of using SWITCHES in SIMULINK typically use a cross-fading technique to smoothly transition between two filters. The cross-fading technique works by gradually increasing the gain of the new filter and decreasing the gain of the old filter over a period of time but does not achieve instantaneous output matching in the subsequent time step, as the proposed method does. To attenuate the transient response during transition from one filter to another based on the gear status, an approach was employed to match the response of the previous filter with the initial condition of the current filter (the filter to which switching is made). This was done to mitigate any abrupt changes and ensure a smoother and more seamless transition between the filters.

The mathematical principles underlying the process are demonstrated as follows.

Consider two discrete time domain filters given:

$$G_1 = \frac{b_1 z}{z + a_1} \text{ and } G_2 = \frac{b_2 z}{z + a_2}$$

Now considering, $G_1 = \frac{b_1 z}{z + a_1}$

$$\Rightarrow \frac{y(z)}{u(z)} = \frac{b_1 z}{z + a_1} = \frac{b_1}{1 + a_1 z^{-1}}$$

$$\Rightarrow y(z)(1 + a_1 z^{-1}) = b_1 u(z) \tag{2}$$

Now taking the inverse '$z$' transform of Equation (2), we get.

$$y_k = b_1 u_k - a_1 y_{k-1}$$

Applying the same procedure for filter 2, we get:

$$y'_k = b_2 u_k - a_2 y'_{k-1} \tag{3}$$

To avoid transience while switching from filter 1 to filter 2:

$$y'_k = y_k$$

$$\Rightarrow b_2 u_k - a_2 y'_{k-1} = b_1 u_k - a_1 y_{k-1}$$

$$\Rightarrow y'_{k-1} = \frac{1}{a_2}((b_2 - b_1)u_k + a_1 y_{k-1}) \tag{4}$$

It is to be noted that the input $u_k$ remains constant for both filters, thereby eliminating the necessity to employ subscripts for the input. Now, substituting Equation (4) in Equation (3) yields the time difference equation to be used while switching from one filter to another.

$$y'_k = b_2 u_k - ((b_2 - b_1)u_k + a_1 y_{k-1}) \tag{5}$$

Equation (5) is used at the time step 't' when the filter switches to ensure smooth transitions between filters during gear changes. It establishes the output from the preceding time step, $y'_{k-1}$, which yields an equivalent output at the current time step, $y'_k$, for the newly engaged filter.

The effectiveness of the zero transient filtering can be seen in Figure 3.

In the context of the above Figure 3 and the interpretation of its results, a TruckSim® simulation was first run using the given series of step throttle inputs. The resulting total drive torque, obtained from the simulation, was subsequently utilized as input to the inverse throttle dynamics filters. The objective was to compare the filter outputs, which, in theory, should correspond identically to the prescribed series of step inputs. The application of zero transient filtering implementation demonstrated significantly reduced transience in its outputs (throttle) in comparison to the conventional implementation that utilizes switching logic, as illustrated.

The validity and reliability of the estimated inverse throttle dynamics transfer function were assessed through a series of comprehensive validation simulation runs. One of the results is depicted in Figure 4. The primary objective of the validation process was to evaluate how well the linear estimated model was able to replicate the non-linear relationship between the throttle and the realized drive torque.

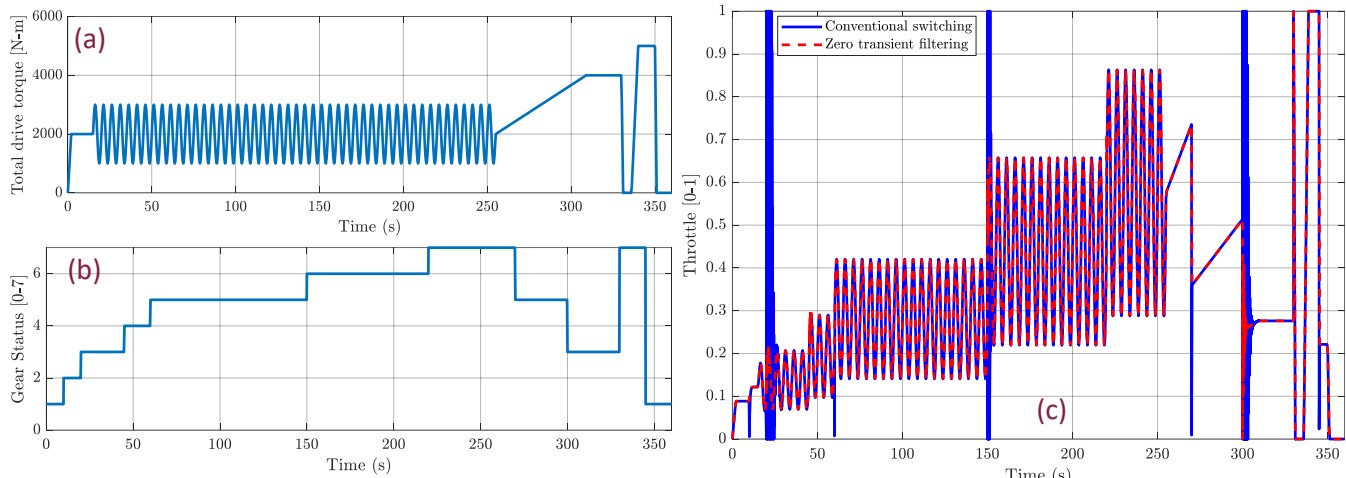

**Figure 3.** Comparison between normal gain scheduling implementation and zero transient filtering (**a**), (**b**) Inputs to filter (**c**) Outputs from filter.

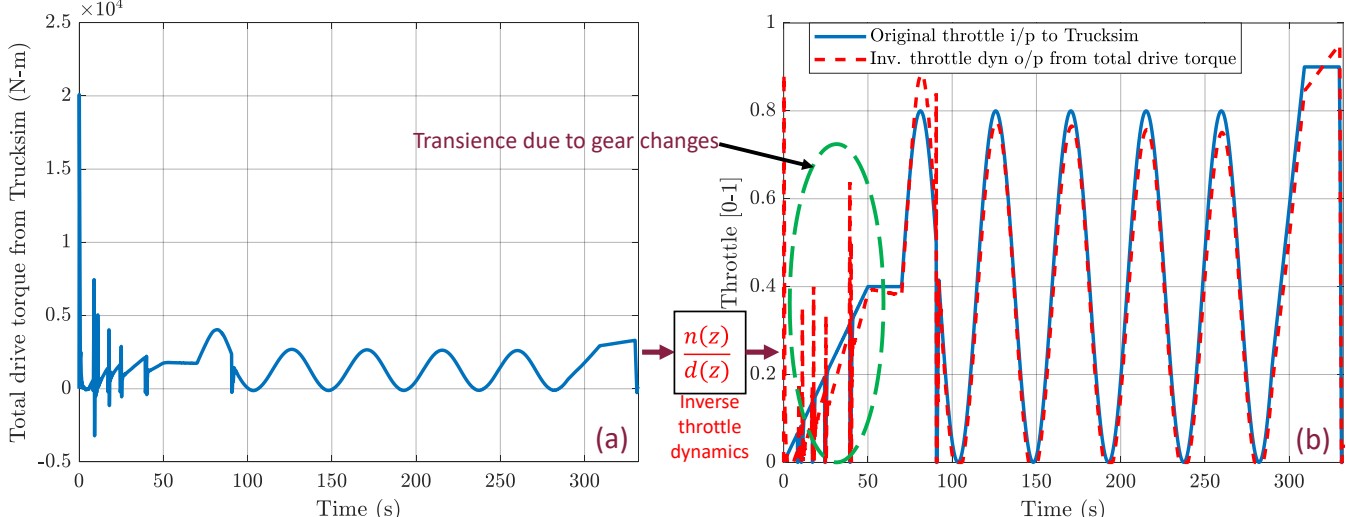

**Figure 4.** Inverse throttle dynamics validation results (**a**) I/p to inverse model (**b**) Estimated throttle vs. Original commanded throttle.

Regarding the methodology (used for validation), a user-defined input representing a combination of step changes, sinusoidal variations, and ramp profiles was provided to the TruckSim® simulation environment. The total drive torque output during the simulation was recorded as a response to this input. Subsequently, the measured drive torque was used as the input to the estimated inverse throttle dynamic model, with the objective of reconstructing the original throttle profile that was initially applied in the TruckSim® simulation.

As depicted in Figure 4a, the total drive torque output from the TruckSim® simulation served as the input to the inverse model, leading to the successful regeneration of the original throttle profile, as demonstrated in Figure 4b. Remarkably, there is a notable agreement between the original throttle profile and the back-estimated throttle profile obtained through the inverse model. It is to be noted that the presence of sudden transient spikes in the throttle profile can be attributed to gear changes, signifying certain complexities inherent to the ICE powertrain system in general.

**Characterization and validation of inverse brake dynamics**

Using linear system ID for characterizing the inverse relationship between the total brake torque required at all wheels and the brake control pressure (one of the ways to

control brake actuation in TruckSim® using Simulink®) presents significant challenges. This difficulty primarily arises from the non-linear relationship existing between the actuating brake pressure and realized brake torque at the wheels. The activation of ABS is one of the reasons. Also, since the brakes are applied to all wheels, the total torque that can be commanded is very high and is only limited by traction.

To proceed with the system ID, different magnitude step inputs, as shown in Figure 5 (Brake control pressure), were fed into TruckSim® as inputs, and the corresponding total brake torque at the wheels was recorded. This combination of input–output data was then used along with MATLAB system ID tool to obtain an optimized '*z*' domain transfer function. This transfer function serves as a mapping mechanism, enabling the association of the total brake torque with the corresponding brake control pressure required for effective brake actuation control in the TruckSim® simulation environment.

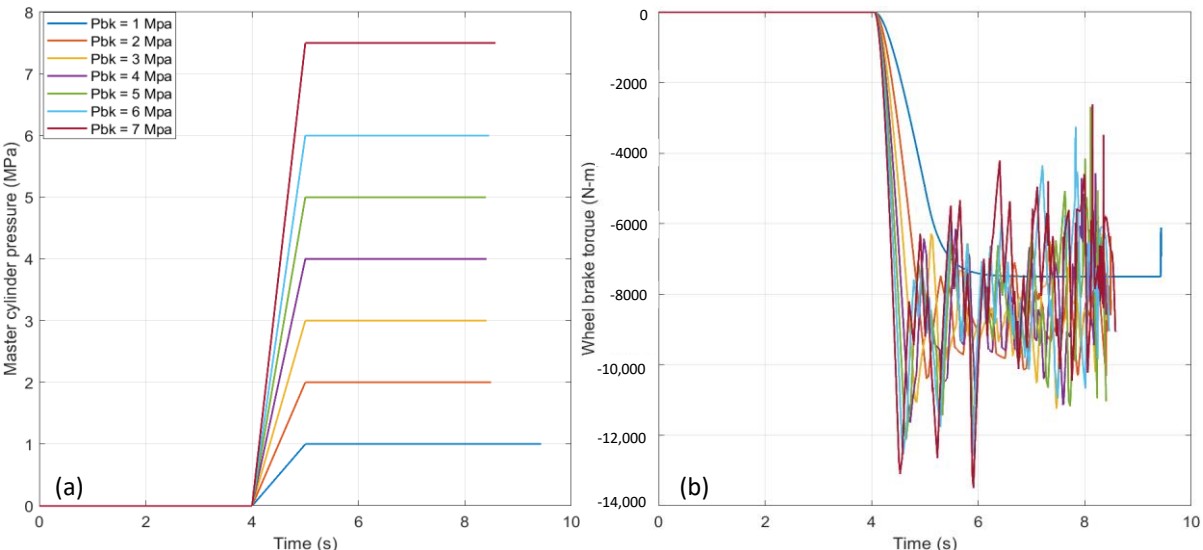

**Figure 5.** Characterizing inverse brake model (**a**) Inputs to TruckSim® as brake control pressure (MPa) (**b**) Corresponding outputs of total brake torque realized.

The analysis of Figure 5 highlights a significant variability in the brake torque output experienced at the wheels in response to changes in the input, specifically brake pressure. As suggested earlier, this is mainly due to the activation of ABS when the brake force at the tire exceeds the traction limit, and the tire starts sliding. In addition to the ABS-induced fluctuations, there are also variations in rise times and the manifestation of saturation effects, resulting in relatively constant steady-state values for different brake pressure inputs, thus indicating the presence of non-linearity in the system.

The validation process for the inverse brake dynamic model followed a methodology like the one used earlier. Figure 6b illustrates the output of the inverse brake transfer function, designed to reconstruct the original actuation brake control pressure input provided to the TruckSim® environment. It is evident from the results that the linear estimated inverse model does not fully capture the intricate non-linear brake dynamics inherent within the TruckSim® simulation.

However, it is important to note that this discrepancy will be compensated upon integration with the closed-loop feedback control system. The closed-loop feedback mechanism compensates for the limitations of the linear model and enhances the overall performance and accuracy in the control of the brake dynamics, as shown later in the longitudinal control results subsection.

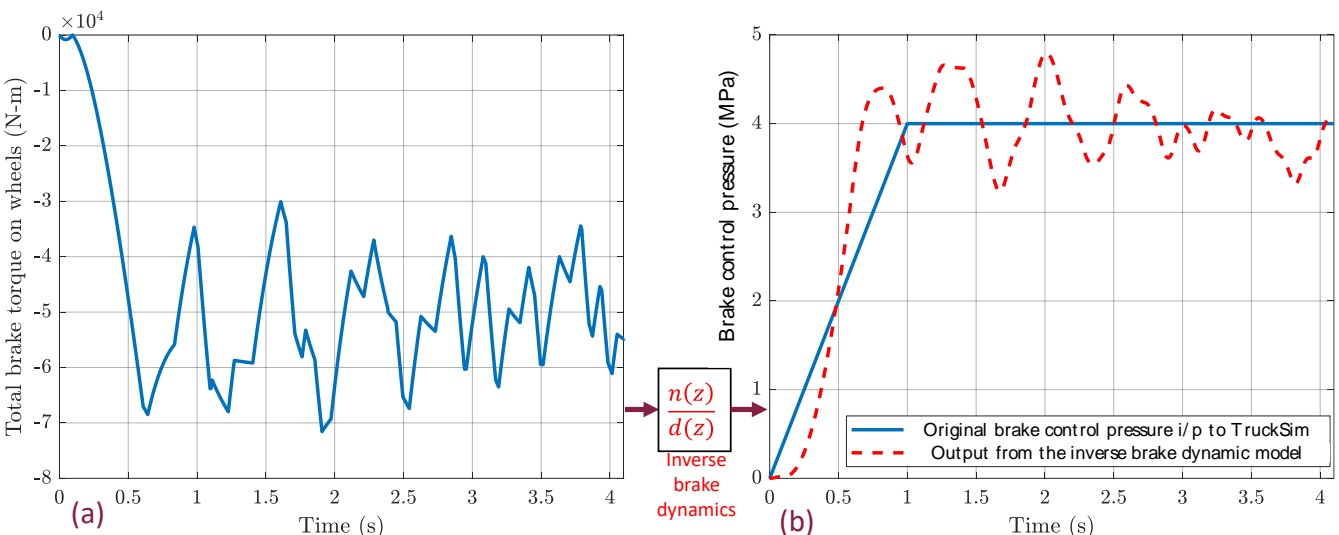

**Figure 6.** Inverse brake dynamics validation results (**a**) I/p to inverse brake model (**b**) Estimated brake control pressure vs. original commanded actuation pressure.

## 3. Controller Design

### Estimation of the longitudinal dynamic model used for controller tuning

The fundamental equation governing longitudinal motion obtained by considering the wheel forces along with aerodynamic and rolling resistance is given below. For more detailed descriptions of longitudinal vehicle dynamics models, refer to [38,39]

$$F_x = F_{drag} + (m_1 + m_2)(a + gsin\phi) + (M + N * v_x)(m_1 + m_2)gcos\phi \tag{6}$$

where,

$F_x$—Longitudinal(brake/traction) force realized at the tyre‾road interfaces;
$F_{drag}$—Aerodynamic drag force acting on the tractor‾trailer;
$m_1$—Mass of the tractor, $m_2$‾Mass of the trailer;
$a$—Longitudinal acceleration of the tractor‾trailer;
$v_x$—Longitudinal velocity of the tractor‾trailer;
$g$—acceleration due to gravity, $\phi$‾Road grade angle;
$M$—constant component of the rolling resistance coefficient;
$N$—Speed varying component of the rolling resistance coefficient.
Excluding the terms dependent on grade ($\phi$) since those can be feedforwarded, we get,

$$F_x = k_1 v_x^2 + k_2(\dot{v}_x) + (M + Nv_x)k_2g \tag{7}$$

where, $k_1 = 0.5C_D A\rho$, $k_2 = m_1 + m_2$.

Linearizing Equation (7) and treating $v_x$ as the output and $F_x$ as the input, the corresponding transfer function that establishes the relationship between the two variables can be derived and is given by:

$$\frac{Vx(s)}{Fx(s)} = \frac{1}{k_2 s + c} \; where \; c = (2k_1 x_0 + Nk_2 gcos\phi)$$

The above relation suggests that the longitudinal dynamics can be adequately approximated by a first-order transfer function.

Using this conclusion of a first-order transfer function being a reasonable approximation to the non-linear longitudinal dynamics, once again, linear system ID was used to get the optimal value of parameters '$k_2$', and '$c$'.

It is to be noted that in practical applications involving longitudinal dynamics estimation, the input to the Engine Management System (EMS) of the tractor can be the

commanded torque, communicated through the Controller Area Network (CAN) bus. This commanded torque along with the recorded longitudinal velocity of the tractor corresponding to the given input drive torque, can then be used for linear system identification of the longitudinal dynamics. However, it is worth mentioning that the TruckSim® simulation environment imposes certain limitations, preventing the utilization of commanded wheel torque as an input. To navigate this constraint, a viable approach was adopted: employing engine torque as the input, a permissible input source.

To derive the transfer function that establishes the relationship between longitudinal force and longitudinal velocity, it is essential to exercise comprehensive control over the input variable, i.e., the longitudinal force (since system ID requires this i/o dataset). To achieve this requisite control over the input, a co-simulation environment was established, integrating both TruckSim® and Simulink®. The longitudinal force was first converted into the total drive torque using the equivalent radius (gain factor). This resultant total drive torque was subsequently translated into the necessary engine torque, effecting a multi-step transformation process. It is to be noted that the required engine torque corresponding to each gear to achieve the stipulated commanded drive torque necessitated the incorporation of gear ratios. To facilitate this in real-time simulation, the "*GearStatus*" variable was fedback from TruckSim®.

**Validating the estimated longitudinal dynamics model**

A linear system identification methodology was again used to derive a continuous time domain transfer function relating the longitudinal force to longitudinal velocity. The model was then validated against TruckSim® by comparing the longitudinal velocity generated in response to the inputs commanded. Further validation was also done by comparing a coast-down test between the linear model and TruckSim®. The graphical representation of the achieved results in Figure 7 attests to a substantiated degree of reliability in the estimated linear model.

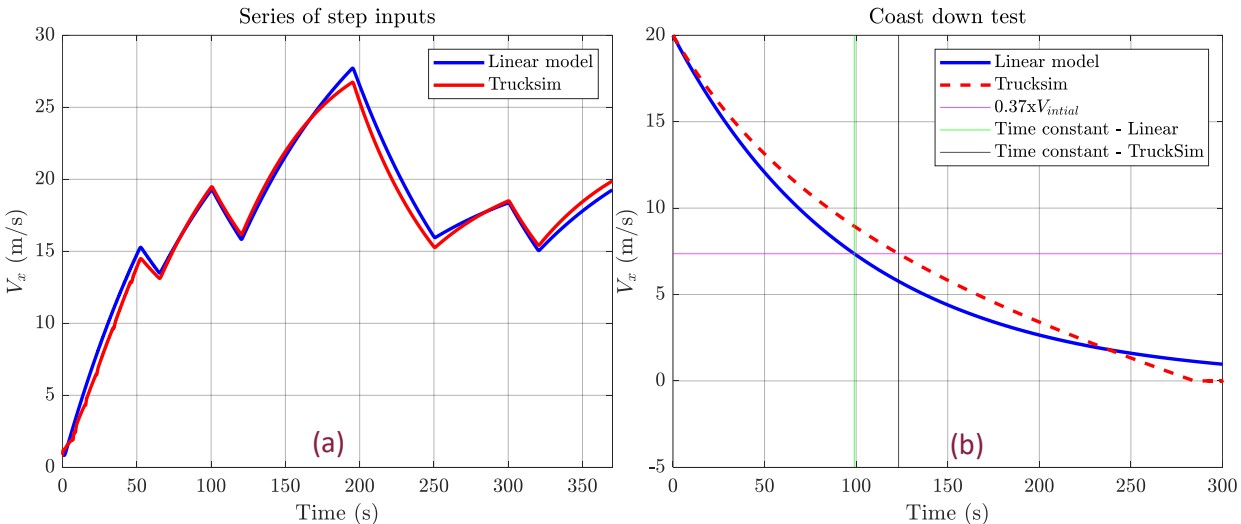

**Figure 7.** Validation of the estimated model (**a**) Velocity response to the commanded $F_x$ (**b**) Coast-down velocity response comparison.

The estimated linear longitudinal model is given by:

$$G(s) = \frac{7.445e - 5}{s + 0.0101} \tag{8}$$

**Designing a nested linear controller for longitudinal control**

The controller design comprises designing a velocity controller (inner loop) and a distance controller (outer loop). The primary objective of the longitudinal controller

implemented on the following vehicles within a platoon configuration is to ensure a safe separation distance (varying or constant depending upon the spacing policy used) from the lead vehicle. The upcoming analysis uses the previously derived linear longitudinal model for tuning both the velocity and distance controllers. Since the transfer function obtained relates longitudinal force to longitudinal velocity, the initial focus involves designing the velocity controller, which aims to drive the relative velocity between the lead vehicle and the following vehicle (within a platoon) to zero. Figure 8 depicts the proposed nested controller structure, which will be used to design both the velocity and the distance controller. As shown the aim of the compensator will be to drive the errors $\Delta V_x$ and $\Delta X$ to zero as quickly and efficiently as possible. It is to be noted that during the loop-shaping process, the variables $V_{xlead}$ and $X_{f_{i-1}}$ will be treated as external disturbances.

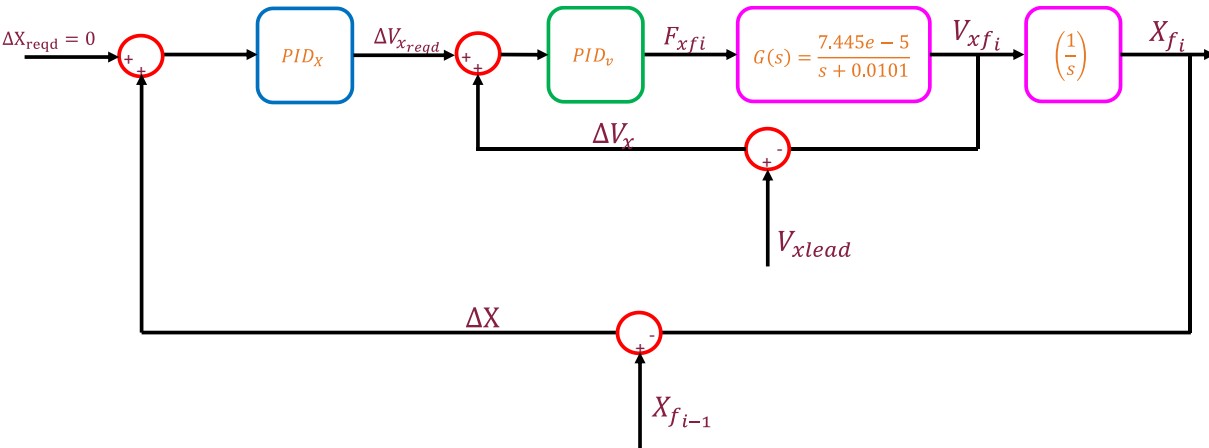

**Figure 8.** The proposed longitudinal controller structure used for tuning both the PIDs.

**Tuning PID for velocity control**

The linear velocity compensator was designed using loop-shaping techniques within the **ControlSystemDesigner** app in MATLAB. The problem has been formulated by specifying the plant function as $G(s) = \frac{7.445e-5}{s+0.0101}$ and developing a compensator for this plant to increase its closed-loop bandwidth while at the same time keeping the actuator efforts needed in check. It is to be noted that the velocity loop will be tuned without considering the variable $V_{xlead}$, primarily because this input is unknown when just the following vehicle is considered. In effect, the tuning process is structured to achieve optimal tracking of a predetermined reference velocity. In this context, the designed controller framework treats $V_{xlead}$ as an external disturbance having the same frequency characteristics as the dynamic model $G(s)$ (which is a valid assumption since $V_{xlead}$ will be the velocity of the lead truck). Within such scenarios, the traditional objective is to achieve a minimal magnitude for the sensitivity function $S = (I + L)^{-1}$, *where* $L(loop\ transfer\ function) = PID_v * G$ across the frequency spectrum of interest for disturbance (in this case $V_{xlead}$). Refer [40].

A comparison of the final sensitivity functions of both controllers has been presented later in Figure 9.

The "PI(D) compensator with a lead filter" designed using loop-shaping techniques to increase the closed loop bandwidth of the linear longitudinal dynamics is given by:

$$C_{velocity} = 11805 + \left(\frac{69.957}{s}\right) - \left(3305 * \frac{3.572s}{(s+3.572)}\right) \tag{9}$$

It is to be noted that in the block diagram shown above (Figure 8), the output of $PID_v$ is the total longitudinal force required, aligning with $G(s)$, which correlates longitudinal force with longitudinal velocity, as established in the system identification section. However, in the final implementation phase (as shown in Figure 1), the controller includes a gain factor

corresponding to the equivalent radius $r_e = 0.51$ m to get the required torque, which is then passed to the inverse dynamics model.

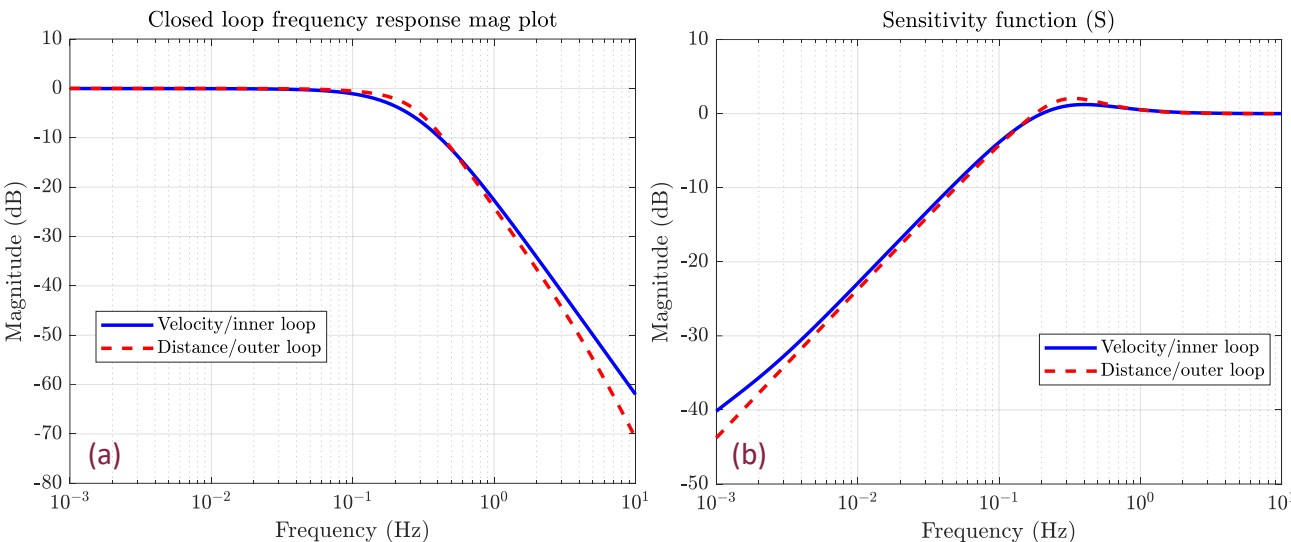

**Figure 9.** (**a**) Closed loop frequency response (magnitude plot) of both the loops (**b**) Sensitivity function magnitude plot of both the loops.

**Tuning linear distance controller**

Once the PID gains of the velocity controller were finalized, block diagram reduction rules were applied to get the velocity-to-position transfer function $G_{open_{dis}} = \frac{G * PID_v}{s(G * PID_v + 1)}$ (note that this transfer function relates $\Delta V_{x_{reqd}}$ to $X_f$), which is then subsequently used to design the outer loop/distance controller.

Again, in this case, as previously mentioned, due to the inherent uncertainty surrounding the variable $X_{f_{i-1}}$, it will be regarded as a disturbance. Subsequently, the $PID_x$ controller was then tuned to achieve tracking of diverse reference distance profiles used as inputs. As expected, the tuning yielded the intended configuration of the sensitivity transfer function in the frequency domain, as shown in Figure 9b. It is pertinent to observe that the velocity-to-position transfer function $G_{open_{dis}}$ within this context inherently features a pole located at the origin, which introduces complexities in tuning the controller. It is noteworthy that the Ziegler-Nichols method was initially employed to derive the initial PID gains. Subsequent refinements in tuning were conducted to optimize the performance of the closed-loop distance controller, arriving at $\mathbf{C_{distance}(s)} = \frac{25.46\mathbf{s} + 30.21}{\mathbf{s} + 13.79}$ in continuous time domain 's' which is equivalent to $\mathbf{C_{distance}(z)} = \frac{25.3\mathbf{z} - 25.27}{\mathbf{z} - 0.9863}$ in discrete time domain 'z'.

The frequency response plots illustrating the closed-loop behaviors of both the velocity and distance loops are presented in Figure 9a. As previously discussed, the sensitivity functions for both loops exhibit a preferred configuration characterized by minimal gains at the frequencies of interest. This desirable behavior is precisely illustrated by the sensitivity magnitude plot exhibiting a decay of 20 dB per decade for frequencies less than ~0.2 Hz.

An interesting observation is that the closed-loop bandwidth of the inner velocity control loop is lower than that of the outer distance control loop. This defies the conventional practice in nested PID control systems, where the inner loop typically exhibits a faster response rate than the outer loop. This outcome emerged from an extensive series of trials involving various PID gain configurations. Interestingly, the optimal results were achieved when the distance control loop exhibited a slightly higher bandwidth than the velocity control loop. It was observed that when the velocity control loop was faster, the performance of the distance control became suboptimal, characterized by a prolonged convergence time of the spacing error to zero. Given the context of platooning, where precise and faster intervehicle distance control holds paramount significance compared to

relative velocity control (although both are interrelated), the selection of PID gains favoring a faster outer loop than the inner loop aligns with the desired control objectives.

**Brief discussion on controller stability**

The focus of this study was to rigorously test the proposed nested PID controller through simulations, which is a crucial step before any practical deployment in automated vehicle platooning systems. Recognizing the significance of stability in such control systems, we present a discussion on the internal and robust stability of the controller, acknowledging that our analysis is not an exhaustive mathematical proof but rather an assurance based on extensive simulation results.

**Internal Stability**

The internal stability of the system was first verified by analyzing the location of the closed-loop poles $L_{outer} = \left( \frac{PID_v\,PID_x\,k}{c\,s + s^2 + PID_v\,PID_x\,k + PID_v\,k\,s} \right)$. All poles have negative real parts, which is a fundamental criterion for internal stability in a control system. The negative real parts of all poles confirm that the system is BIBO stable. This means that for any bounded input, the system's output will remain bounded, which is a crucial characteristic for ensuring safe and predictable control of vehicle platoons.

Also, the Bode plot analysis of the overall closed loop $L_{outer}$ yielded the following results:

- Gain Margin: 5.7 (The system can tolerate a gain increase of up to 5.7 times before becoming unstable).
- Phase Margin: 63.72 degrees (This significant phase margin indicates a considerable buffer before the system reaches the critical $-180$ degrees phase shift, at which point instability would occur).

**Analysis of Robust Stability**

In the evaluation of the nested PID control architecture, robustness analysis plays a critical role, particularly due to the presence of parametric uncertainties and unmodeled dynamics. Uncertainties in the plant $G(s)$ have been introduced in the parameters '**k**' and '**c**', each with a variability of up to 20%. The unmodeled dynamics, which may arise from various sources such as non-linearities or unanticipated interactions within the system, have been encapsulated using the uncertainty block '**Delta**', as shown below and in Figure 10.

$$G_{nom} = \frac{k}{s+c}$$

$$G_{param\ uncertain} = \frac{(k \pm 20\%)}{s + (c \pm 20\%)}$$

$$G_{uncertain} = G_{param\ uncertain}(1 + \delta * w), \ where \ w = 0.4 \pm 20\%$$

where, $G_{nom}$ is the nominal plant transfer function.

$G_{param\ uncertain}$ is the plant transfer function with parametric uncertainty.

$G_{uncertain}$ is the overall uncertain plant transfer function incorporating both the parametric and dynamic uncertainty.

The robust stability of the system was quantified using a $\mu$-analysis, which revealed a system capable of withstanding up to 109% of the modeled uncertainty. A destabilizing perturbation was identified at 110% of the modeled uncertainty, which could induce instability at a frequency of 2.34 rad/seconds = 0.37 Hz. In the context of an automated vehicle platoon, a destabilizing perturbation could manifest in (a) **Vehicle Dynamics Variations:** Differences in the dynamics of each vehicle, which may not be captured in the nominal model, can act as a perturbation. For example, variations in vehicle mass, tire characteristics, or suspension settings due to load changes, tire wear, or different vehicle maintenance states; (b) **Actuation System Variations:** Differences in the performance of the actuation systems (like throttle or brake response times) between vehicles can create perturbations. If one vehicle's brakes respond slower than expected, it could potentially destabilize the platoon; (c) **Communication Delays:** In a platoon, vehicles communicate to maintain tight

formation. Any variation in the communication delay could be a perturbation. For example, if a vehicle suddenly starts experiencing a longer delay in receiving signals, it could disrupt the coordination, etc.

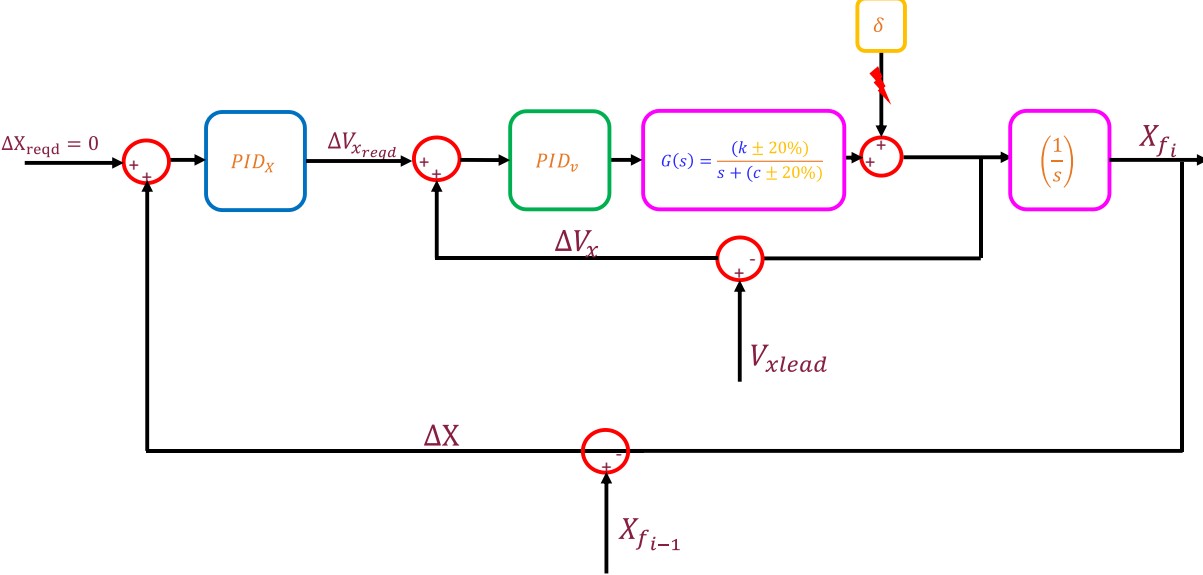

**Figure 10.** Block diagram representation of the modeled uncertainty.

The sensitivity analysis component of the robustness evaluation provided insights into which uncertain elements have the most significant impact on the stability margins. The results indicated that the uncertainty block $\delta$, representing unmodeled dynamics, had the highest influence, with an 84% contribution to the overall margin. A 25% increase in $\delta$ would lead to a 21% decrease in the stability margin. Conversely, the parameters c and k exhibited considerably less influence, with c showing no impact on the margin and k accounting for an 11% contribution.

It is essential to clarify that the primary contribution of this work is the development and simulation-based validation of a control strategy for vehicle platooning. While our stability discussion is not comprehensive, it provides foundational insights into the behavior of the system. The internal stability and μ-analysis suggest that the proposed controller is a promising candidate for further investigation and practical application. We conclude this section by asserting that the pursuit of a formal string stability proof is a critical next step for this line of research.

## 4. Results and Discussion

This section presents the conclusive outcomes concerning the longitudinal control of tractor–trailers within an automated platoon. To streamline the results and align with the pragmatic considerations related to the platooning of such heavy vehicles, the analysis has been confined to scenarios involving a platoon of three vehicles (refer to Figure 11a). It is important to note that while expanding platoon size has been explored, practical feasibility and potential drawbacks, such as the risk of bridge overloading due to multiple fully loaded tractor–trailer combinations, have influenced this simplification. The vehicle used in this simulation is a 2A Day cab tractor (225 kW) with 22 feet trailer (capacity—10 tons).

The procedure commenced by subjecting the lead vehicle to distinct simulation trials, each characterized by diverse throttle profile inputs. This was done to assess the efficacy of the implemented longitudinal controller in maintaining desired tracking performance. Furthermore, the robustness of the controller was also evaluated by introducing road grade changes and by varying payloads within the trailing/following vehicles. These simulation runs were aimed to determine how diverse operating conditions impact the controller performance.

The quantitative analysis of the tracking performance was accomplished by examining spacing errors and tracked velocities within the platoon, as discussed in the subsequent subsections.

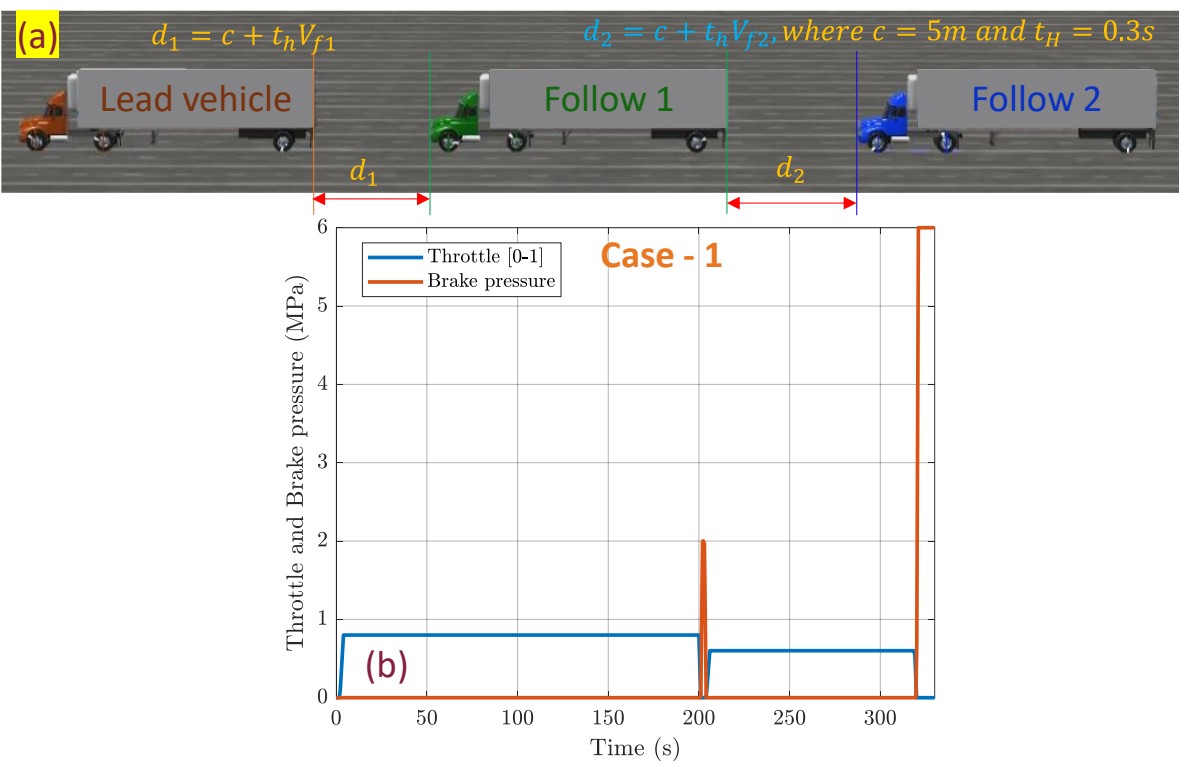

**Figure 11.** (**a**) Three vehicle platoon (**b**) Step rise throttle and m brake control pressure inputs to the lead vehicle.

**Step throttle input to lead vehicle**

Figure 11 illustrates the control inputs (throttle as well as the master cylinder brake pressure) governing the velocity and acceleration of the lead vehicle. The throttle input consists of a step function, with an initial rise from 0 to 0.8 executed approximately 2 s into the simulation. This level is sustained until the 240 s mark, after which it reverts to zero. During instances of zero throttle, a swift brake impulse is introduced, as demonstrated. This strategic maneuver aims to assess the ability of the longitudinal controller to reduce the spacing error to zero while also testing its efficacy in mitigating potential collisions in emergency situations by effectively inducing deceleration. Notably, the road elevation was maintained at a constant value for this simulation.

The outcomes of the longitudinal control simulations are graphically presented in Figure 12. The spacing error time history shown in Figure 12a clearly demonstrates the good tracking capability of the implemented controller. This ensures that the following trucks can follow the leading truck smoothly while consistently maintaining a safe intervehicle distance. Figure 12b, which shows the tracked velocity profile, proves that the following trucks can track the velocity of the leader rapidly and smoothly. It is to be noted that there exists a steady state error of $\sim$ 0.2 m between successive vehicles within the platoon. This disparity, while present, remains insignificant when compared to the minimum intervehicle distance of 5 m.

Furthermore, another interesting observation pertains to scenarios where the trailer's payload approaches its maximum capacity. It is noteworthy that, under such circumstances, the following vehicle with a higher payload necessitates more time to minimize the intervehicle distance due to actuation constraints imposed by limited engine torque output. However, once the spacing error approaches negligible levels, subsequent simulation peri-

ods demonstrate consistent spacing error maintenance well within secure limits, further affirming the controller's effectiveness and robustness.

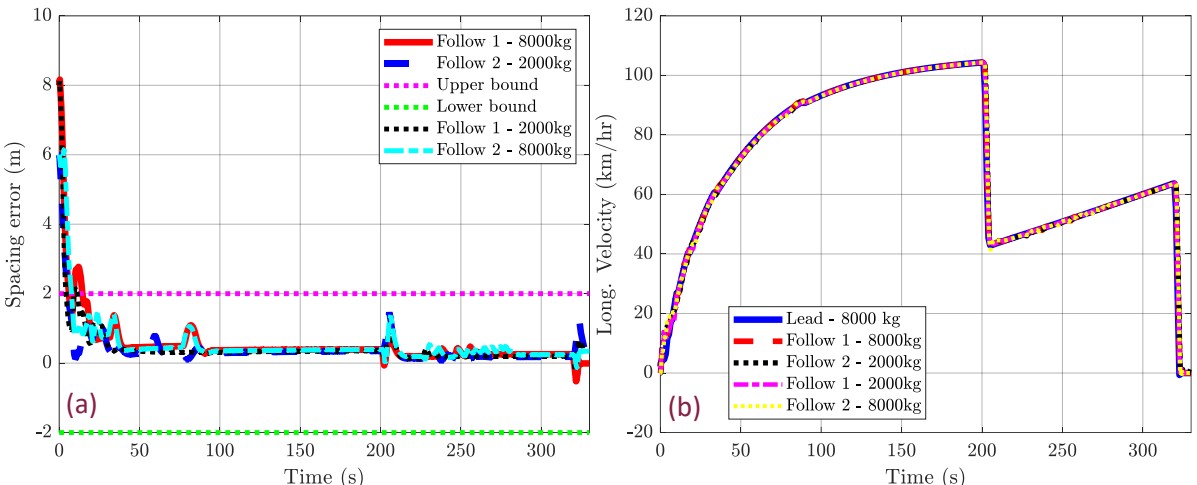

**Figure 12.** (**a**) Spacing error vs. time for both following vehicles (**b**) Follow vehicles' velocity time history.

**Ramp throttle input to lead vehicle**

In this case, as shown in Figure 13a, the throttle was slowly ramped up and then ramped down to demonstrate the combined ramp and step response of the platooning vehicles. In this case, a considerable amount of elevation change was imposed over the entire path to test the robustness. The results show that spacing error remains contained within a safety margin of $\pm 2$ m. This value of the safety threshold was influenced by a recent experimental study on truck platooning [26], where the root mean square (RMS) spacing error was measured to be 2.3 m, utilizing an MPC approach for longitudinal control. (Figure 14a). This deviation is attributed to the abrupt reduction in drive torque experienced by the trailing vehicles during gear shifts. Nonetheless, it is worth noting that even with elevation changes and different payloads on each following vehicle, the maximum extent of this error remains bounded within approximately $\pm 2$ m.

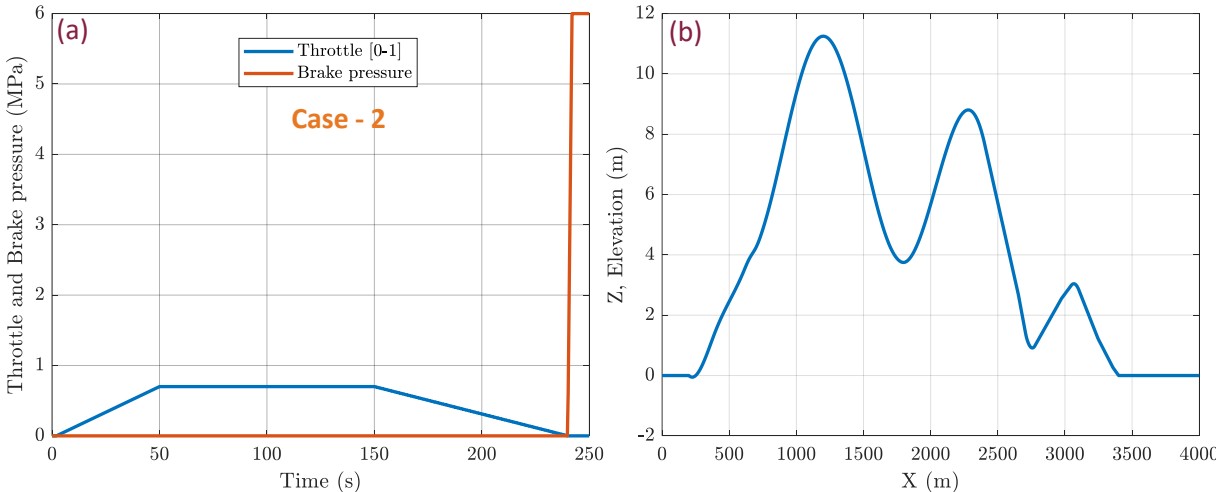

**Figure 13.** (**a**) Ramp-shaped throttle and brake control pressure inputs to the lead vehicle (**b**) Elevation (Z) vs. Station (X) plot.

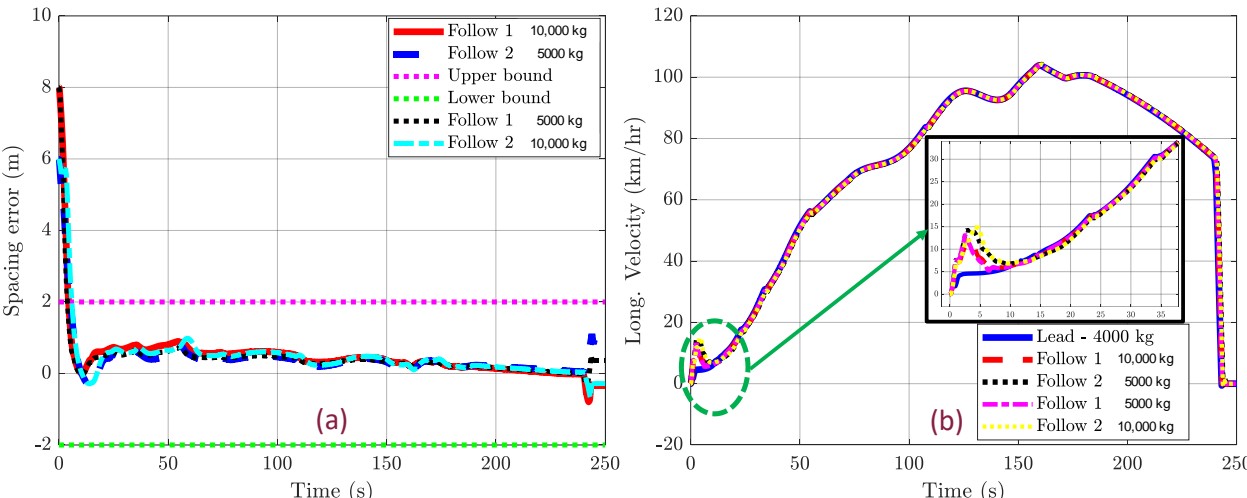

**Figure 14.** (**a**) Spacing error vs. time for both following vehicles (**b**) Follow vehicles' velocity time history.

### Custom throttle input to lead vehicle

In this simulation, the controller was tested by subjecting the lead vehicle to a custom combination of inputs, incorporating step, ramp, and sinusoidal components, as illustrated in Figure 15. The associated spacing error and relative velocities are illustrated in Figure 16. Notably, the spacing error remains confined within a secure margin of $\pm 2$ m.

Once again, variation in terrain elevation was introduced to assess the controller's ability to realistically maintain tracking. It is worth highlighting that the frequency of the sinusoidal throttle variation remains well below the overall cutoff frequency, which encompasses both powertrain and longitudinal dynamics. This is evident in the velocity plots which clearly show the corresponding sinusoidal fluctuations. One minor drawback that merits attention is the relatively larger spacing error observed in the first following vehicle when the entire platoon comes to a sudden stop. Nevertheless, this error does not exceed 2 m even when the payload is close to its maximum capacity. Additionally, it is crucial to mention that the abrupt braking maneuver was deliberately designed to test the system in one of the worst-case scenarios.

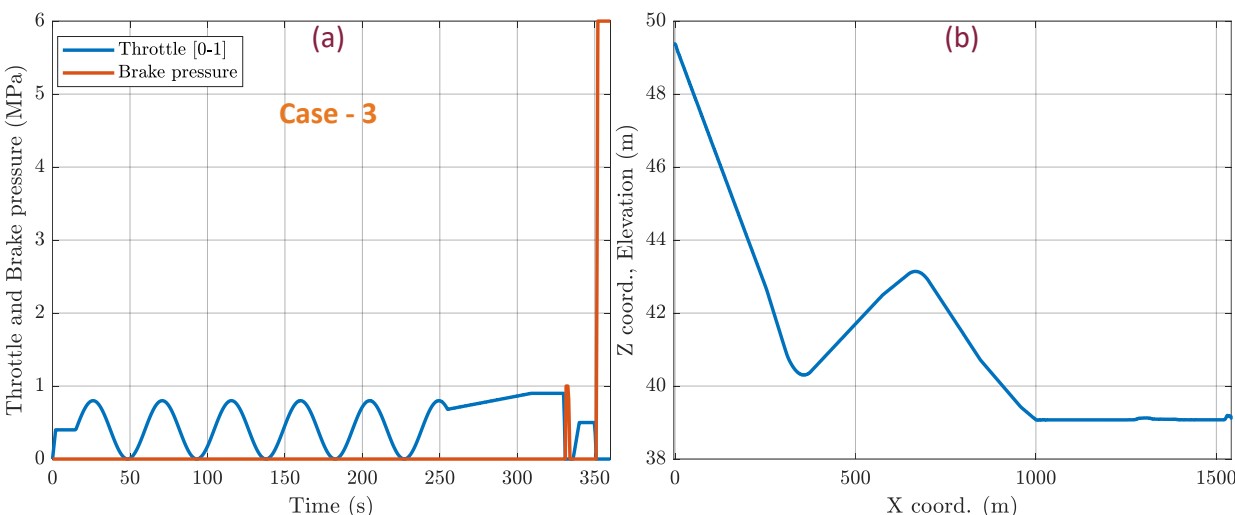

**Figure 15.** (**a**) Customized profile throttle and brake control pressure inputs to the lead vehicle (**b**) Elevation (Z) vs. Station (X) plot.

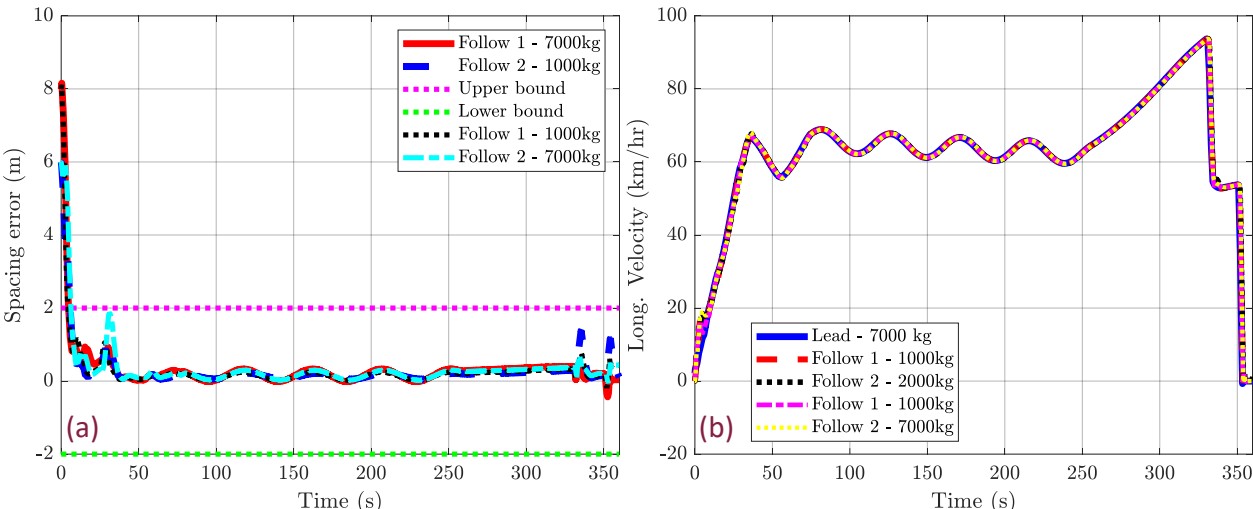

**Figure 16.** (**a**) Spacing error vs. time for both following vehicles (**b**) Follow vehicles' velocity time history for custom combination of inputs.

Another noteworthy aspect in all three simulation case studies is the deliberate initialization of simulations with a non-zero spacing error. This scenario, which is less commonly addressed in existing literature to the best knowledge of the authors, adds an additional layer of realism and practicality to the presented study.

**Comparative analysis with an existing control strategy**

In this subsection, results of a comparative analysis between the proposed PID controller and an existing non-linear controller based on sliding mode control, as detailed in [16], have been presented. The evaluation demonstrated that the PID controller is on par with, or in some cases, outperforms the SMC, particularly in managing the spacing error for the second following vehicle in a platoon. As shown in Figure 17, the spacing error for the PID controller offers improved tracking control as compared to the SMC. It is also important to note that the comparative study used an analytical vehicle model supplemented by a transport lag equation for modeling the actuation delays. While this model is comprehensive, the use of the TruckSim vehicle model, which is validated by empirical real-world data, offers a more accurate representation of vehicle dynamics. The fact that the presented PID controller achieves comparable results using this more sophisticated and validated model underscores the robustness and efficacy of the proposed control strategy.

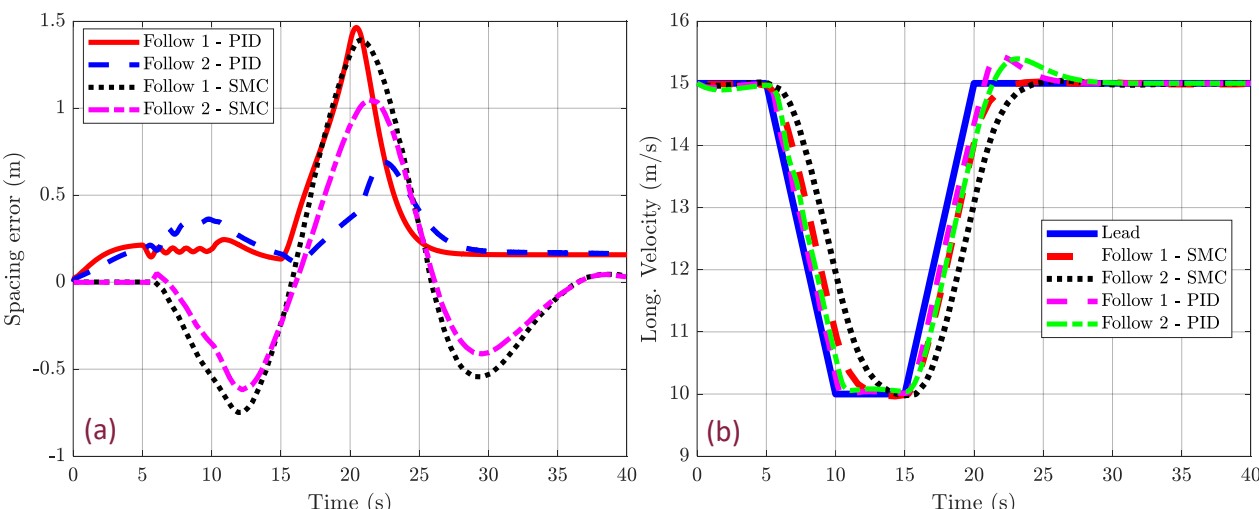

**Figure 17.** Comparative results with an existing non-linear controller (**a**) Spacing error vs. time (**b**) Follow vehicles' velocity time history.

**Discussion on controller effort**

To compare the platooning performance, a comparative analysis of the controller effort in the context of the longitudinal control of an automated platoon consisting of three vehicles has been conducted. The controller output, which in this case is the total torque commanded at the wheels, has been evaluated for both following vehicles in relation to the lead vehicle in both Figures 18a and 19a. Additionally, the total fuel consumed over each simulation run has also been examined and plotted in Figures 18b and 19b. It is to be noted that in this analysis, the energy advantage platooning offers in terms of reduced aerodynamic resistance has not been considered. Rather, the aim of the study is to check the aggressiveness and abruptness of the commanded controls. To ensure a fair comparison, the fuel consumption subroutine used in all three vehicles was the same. Also, compared to the previous simulation runs, nothing else was changed other than making the payload the same at 4000 kg.

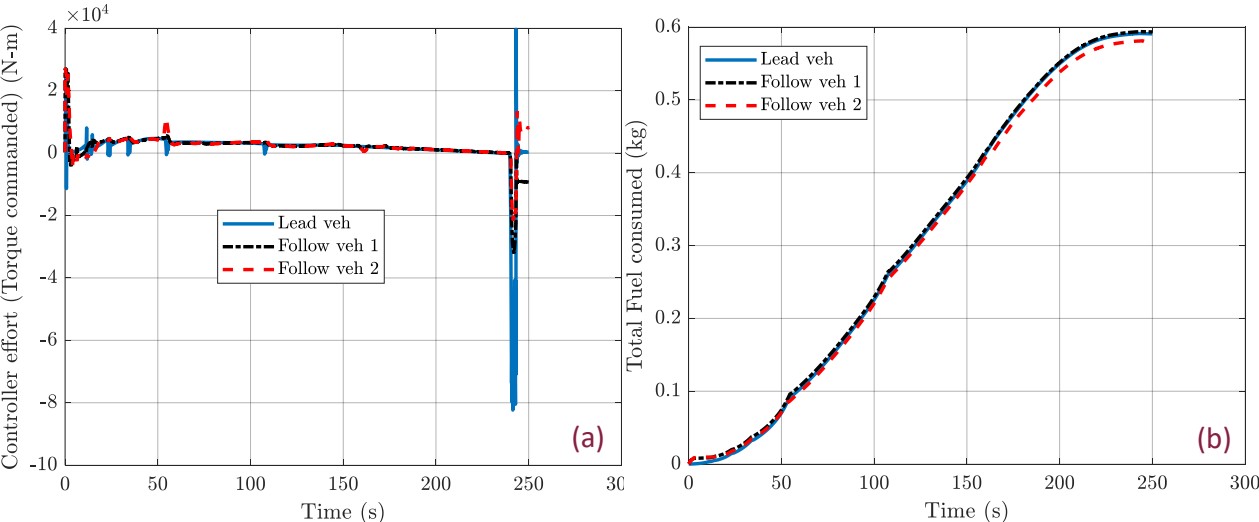

**Figure 18.** Comparison of controller effort (**a**) Total torque realized on wheels (**b**) Total fuel consumed over the entire run for ramp response.

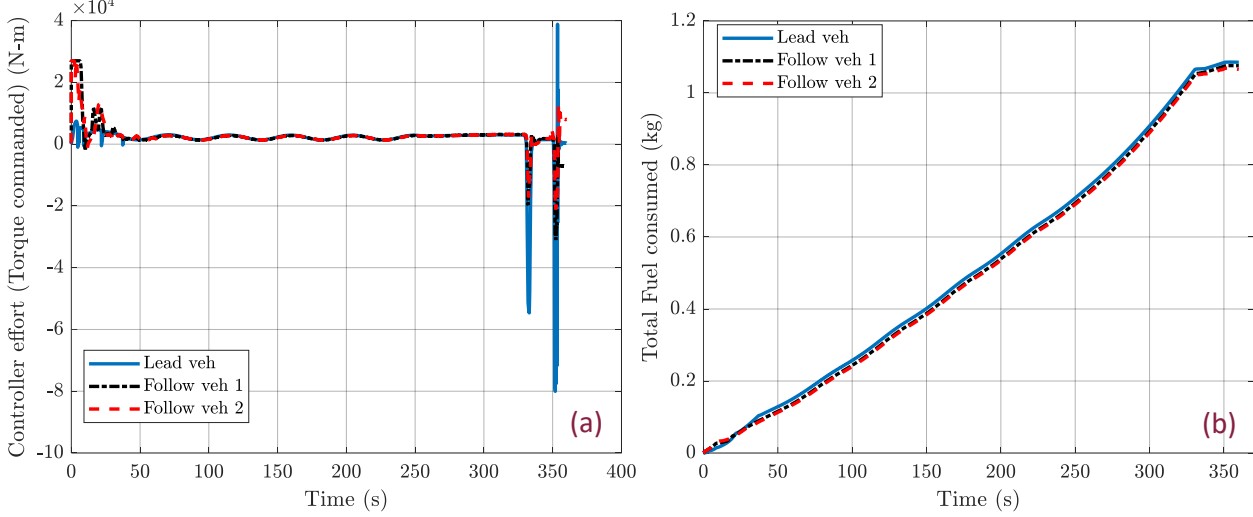

**Figure 19.** Comparison of controller effort (**a**) Total torque realized on wheels (**b**) Total fuel consumed over the entire run for custom combination of inputs.

Observing Figures 18a and 19a, it can be deduced that the controller effort demonstrated by both following vehicles displays minimal occurrences of aggressive or abrupt commands. Furthermore, across the presented scenarios, it becomes evident that the fuel

consumption of both following vehicles is less than or equal to that of the lead vehicle (Figures 18b and 19b). However, a slight anomaly emerges when considering the fuel consumption of the first following vehicle in the context of the ramp throttle input scenario. In this case, its fuel consumption is slightly higher or on par with that of the lead vehicle. This can be attributed to a relatively larger initial spacing error, which was present at the beginning of the simulation when compared to the initial spacing error of the second following vehicle. This conclusion can also be verified from the fuel consumption plot, where the first following vehicle's fuel consumption surpasses that of the others around the 50-s mark and then follows the same trend. These observations highlight the predictive capability of the implemented control strategy attributed to the utilization of relative velocity between the ego vehicle and the lead vehicle, rather than solely considering the spacing error between the ego and the preceding vehicle.

## 5. Conclusions and Future Works

Truck platooning offers the capability of improving road transportation efficiency, diminishing fuel usage, and mitigating carbon emissions. By facilitating coordinated communication among vehicles, platooning has the capacity to optimize traffic dynamics, alleviate congestion, and contribute to the establishment of safer, environmentally friendly, and enduring freight transportation systems.

In this simulation study, the aim was to develop a longitudinal controller for the following vehicles in a platoon and test it in simulation using the TruckSim® vehicle model, which incorporates powertrain non-linearities, actuation delays, and gear changes—features often excluded in prior studies. The implemented controller successfully constrained spacing error within ±2 m and adhered to relative speed constraints. The performance of the controller was tested by simulating the platooning operation under a wide variety of lead vehicle operation cycles (velocity profile). The robustness of the controller was also evaluated by imposing grade changes and by varying the payload masses on both following vehicles to simulate real-world scenarios of varying operating conditions. Comparative analysis of controller effort highlighted the alleviation of conventional PID issues, such as aggressive actions, oscillations, and overshoots, thus improving its applicability.

One potential future direction would be to explore the overall benefits of gain-scheduled PID strategies based on operational conditions, particularly payload. Additional areas of exploration include the inclusion of communication delays, packet data losses, and even the integration of fluid dynamics models to assess aerodynamic gains. Moreover, a thorough theoretical proof of string stability and an analysis of the maximum tolerable communication delay for maintaining stability with the proposed controller are crucial areas for further investigation.

**Author Contributions:** Conceptualization: A.S., S.S. and M.A.; methodology: A.S. and S.S.; software: A.S.; validation, A.S.; formal analysis: A.S., S.S. and M.A.; investigation: A.S. and S.S.; resources: M.A.; data curation, A.S.; writing—original draft: A.S.; preparation, A.S.; writing—review and editing, A.S., S.S. and M.A.; visualization, A.S. and S.S.; supervision: S.S. and M.A.; project administration, M.A.; funding acquisition, M.A. All authors have read and agreed to the published version of the manuscript.

**Funding:** This research received no external funding.

**Data Availability Statement:** The findings of this study are supported by data generated through comprehensive simulation experiments. All relevant data underpinning the conclusions of this research are included within the article. As this is a simulation-based study, there are no raw experimental data sets. However, for further inquiries or additional details regarding the simulation setup and implementation, interested parties are encouraged to contact the first author. The corresponding author is committed to providing any necessary information to facilitate understanding and replication of the study's results.

**Conflicts of Interest:** The authors declare no conflict of interest.

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
