# Peer review of "PID-Based Longitudinal Control of Platooning Trucks"

_machines, doi:10.3390/machines11121069_

Round 1
Reviewer 1 Report
Comments and Suggestions for Authors
This manuscript presents the development and assessment of a PID-based longitudinal control method for platooning trucks. This work has possible research. However, this manuscript has important deficiencies. The main remarks are as follows:
1. Few novelties are shown and the presented work is not solid. The presented PID-based longitudinal control is a conventional method and it is easy to implement for simulation study.
2. The presentation is not well. Section 2 “TruckSim®-Simulink® co-simulation framework” is described in detail. However, this manuscript should focus on the theoretical method rather than the implementation of the simulation system, since this is an academic paper, not a technical document.
3. It is recommended that the Abstract is described in one paragraph instead of multiple paragraphs.
4. There are many misstatements, such as: “The effectiveness of the zero transient filtering can be seen in Figure 3Figure 3.” in line 182 of page 5; “In the context of the above Figure 3Figure 3 and the interpretation of its results” in line 185 of page 5.
Comments on the Quality of English LanguageThe quality of the English language is OK, but the writing of academic papers needs to be improved.
Reviewer 2 Report
Comments and Suggestions for Authors
This study proposes a nested control scheme based on PID control for platooning trucks. The topic is valuable and related to the scope of the journal. However, this paper's motivation and novelty are unclear so far, and the technical description is insufficient. Here are some comments for reference.
1. What is the difference between platooning common vehicles and trucks? The platooning control of vehicles has been well-explored. Why can not these existing methods be applied to trucks?
2. Actually, both CarSim and TruckSim can accept the driven or braking torque of the wheel as the input.
3. How could zero transient filtering take effect during gear switching? (2)-(5) only proves that the output remains the same when switching from one gear to another. This is also true for traditional ways. Therefore, what is the inherent difference between the proposed filter and other methods?
4. The transfer function relates the longitudinal force to the longitudinal velocity in (8), while, as mentioned by the author, the input of the transfer function should be torque in Fig. 8. Please clarify.
5. Most scenarios in simulation tests are acceleration maneuvers. The author should include more braking maneuvers to validate the proposed method holistically.
6. It is recommended that the author compare the existing methods with the proposed one in Section 4.
Reviewer 3 Report
Comments and Suggestions for Authors
The manuscript addresses a current issue, considering that autonomous vehicles will be increasingly present on the streets and highways. The authors focused on the development and assessment of a Proportional, Integral, and Derivative based computationally cost-efficient longitudinal control algorithm for platooning trucks.
The research is presented logically and clearly, being easy to follow. The results are relevant to the field addressed.
The reference section includes articles published in the last five years (more than half), but the rest are also relevant to the researched field. Only the last author has three self-citations.
The manuscript is scientifically sound and the simulations well designed.
The figures are clear and suggestively explain the results obtained.
The conclusions summarize the research conducted.
My specific comments are the following:
Row 62 – “proposed by Cremer D [21],” – Letter D is not necessary (actually is M). Also, please correct at reference section entry 21 which has no author.
Row 182 – “can be seen in Figure 3Figure 3” – please correct.
Row 185 – “Figure 3Figure 3 and the interpretation” – please correct.
Rows 199, 209, 211 – “Figure 4Figure 4” – please correct.
Rows 225, 236 – “Figure 5Figure 5” – please correct.
Row 310 - “Figure 7Figure 7” – please correct.
Row 413 – “Figure 10Figure 10” – please correct.
Row 423 - “Figure 11Figure 11” – please correct.
Row 448 - “Figure 13Figure 13” – please correct.
Rows 491, 493, 506, 510 – “Figure 16 Figure 16(a) and Figure 17 Figure 17(a)” – please correct.
Please review page 17 and remove redundant text.
Round 2
Reviewer 1 Report
Comments and Suggestions for Authors
No more comments.
Comments on the Quality of English LanguageNo more comments.
Reviewer 2 Report
Comments and Suggestions for Authors
Thanks for the author’s responses and explanations. Some of my concerns have been addressed. However, there are issues that need to be further clarified.
1. What is the relationship between the controller mentioned in Fig. 1 and the two PID controllers in Fig.8? Would the PIDx and PIDv act as the controller in Fig.1? If yes, as I mentioned before, the output of PIDv should be the torque, not the force. The output of PIDv should be labeled in Fig.8.
2. How does the author consider the time delay caused by gear shifts, as mentioned in the rebuttal, when identifying G(s)? If the G(S) is not accurate enough, how could the author promise the tuning parameters of PIDx and PIDv are appropriate?
3. The control commands of the following vehicles should be demonstrated in Section 4.
4. I still suggest the author compare the existing method with the proposed method. Because they did not evaluate the challenging scenarios, for example, emergency braking, the merits and potentials of the proposed method could be highlighted.
